# Does LLM Alignment Really Need Diversity? An Empirical Study of Adapting RLVR Methods for Moral Reasoning

## Abstract

Reinforcement learning with verifiable rewards (RLVR) has achieved remarkable success in logical reasoning tasks, yet whether large language model (LLM) alignment requires fundamentally different approaches remains unclear. Given the apparent tolerance for multiple valid responses in moral reasoning, a natural hypothesis is that alignment tasks inherently require diversity-seeking distribution-matching algorithms rather than reward-maximizing policy-based methods. We conduct the first comprehensive empirical study comparing both paradigms on MoReBench. To enable stable RLVR training, we build a rubric-grounded reward pipeline by distilling GPT-5 into a Qwen3-1.7B judge model. Contrary to our hypothesis, we find that distribution-matching approaches do not demonstrate significant advantages over reward-maximizing methods as expected on alignment tasks. Through semantic visualization mapping high-reward responses to semantic space, we demonstrate that moral reasoning exhibits more concentrated high-reward distributions than mathematical reasoning, where diverse solution strategies yield similarly high rewards. This counter-intuitive finding explains why mode-seeking optimization proves equally or more effective for alignment tasks. Our results suggest that alignment tasks do not inherently require diversity-preserving algorithms, and standard reward-maximizing RLVR methods can effectively transfer to moral reasoning without explicit diversity mechanisms.

## 1 Introduction

Recent advances in reinforcement learning with verifiable rewards (RLVR) for large language models (LLMs) have achieved impressive performance in well-defined, structured domains by directly optimizing long context chain-of-thought reasoning (Jaech et al., 2024; Guo et al., 2025; Comanici et al., 2025). However, existing approaches primarily target logical reasoning tasks, especially mathematics (Cobbe et al., 2021) and coding (Chen et al., 2021), leaving their potential in alignment and moral reasoning largely unexplored. Intuitively, alignment tasks typically admit multiple valid answers that reflect different ethical frameworks and value systems, in stark contrast to mathematical and coding problems, which usually have only one objectively correct solution. Therefore, in this paper, we investigate a natural question: ***Is introducing diversity key to adapting the strong reasoning capabilities that RL brings to the logical reasoning into LLMs' alignment and moral reasoning?***

Existing RL methods for LLM reasoning can be broadly categorized into two paradigms. The first category encompasses reward-maximizing methods rooted in PPO (Schulman et al., 2017), which aim to identify an optimal policy that maximizes reward functions under specific regularization constraints. Most current mainstream RLVR methods, including RLHF-style PPO (Schulman et al., 2017; Christiano et al., 2017; Ouyang et al., 2022), GRPO (Shao et al., 2024), and DAPO (Yu et al., 2025), fall into this category and focus on finding a policy mode generally seeking a single high-reward strategy (Li et al., 2025). The second category consists of distribution-matching methods, which learn the flow between policy and reward distributions to enable the policy to capture fine-grained details of the reward landscape. By explicitly modeling this flow, methods like FlowRL (Zhu et al., 2025) can discover diverse solutions and achieve superior performance on complex tasks. Given the differences between these two paradigms, we hypothesize that, compared with

reward-maximizing methods, distribution-matching methods, with the ability to capture diversity, may be more suited for alignment tasks.

To investigate this hypothesis, we conduct a comprehensive empirical study on MoReBench (Chiu et al., 2025), a challenging moral reasoning benchmark that consists of two complementary subtasks: MoReBench-Public, which requires models to reason about value-laden dilemmas in real-world scenarios, and MoReBench-Theory, which tests reasoning consistency under specific philosophical frameworks including utilitarianism, deontology, virtue ethics, care ethics, and justice as fairness. Following the original benchmark's evaluation protocol, we distill GPT-5 (Singh et al., 2025) by training a Qwen3-1.7B-Base model (Yang et al., 2025) to serve as our judge model, which evaluates responses based on detailed rubrics capturing the complex nature of moral reasoning.

Our experiments reveal several surprising findings that challenge our initial hypothesis. First, we observe that reward-maximizing methods can achieve even superior performance compared to distribution-matching methods on moral reasoning tasks. Moreover, through detailed analysis of reward distributions, we demonstrate that alignment rewards are not necessarily more diverse than reasoning tasks in high-reward regions, in most cases, math reasoning tasks exhibit even greater diversity, contrary to the conventional opinion that alignment requires diversity-seeking algorithms. These findings all suggest alignment does not necessarily need to introduce diversity. With sufficiently discriminative verifiable rewards, standard reward-maximizing methods can effectively transfer reasoning capabilities to moral reasoning without explicitly promoting solution diversity.

In summary, our contributions are threefold. **Firstly**, we build a rubric-grounded verifiable reward pipeline for moral reasoning by distilling GPT-5 into a compact Qwen3-1.7B judge, enabling stable reward computation and controlled RLVR training on MoReBench. **Secondly**, we present the first systematic comparison of reward-maximizing and distribution-matching methods on moral reasoning, and show that reward-maximizing methods can match or outperform distribution-matching ones, challenging the view that alignment requires diversity-seeking algorithms. **Lastly**, we analyze reward distributions and demonstrate that high-reward regions in moral reasoning are not inherently more diverse than those in logical reasoning, explaining why standard reward-maximizing methods can transfer reasoning capabilities to moral reasoning without explicitly promoting diversity.

## 2 RELATED WORK

In this section, we will review the relevant literature from two research areas that our study bridges: RL methods for reasoning tasks as well as LLM alignment and moral reasoning. We will elaborate on them separately below.

**RL Methods for LLM Reasoning.** RL post training is widely used to strengthen LLM reasoning. A representative thread is RLHF (Schulman et al., 2017; Christiano et al., 2017; Ouyang et al., 2022), which learns rewards from human preferences and motivates later RL reasoning methods. Under the verifiable reward setting, rewards can be generated automatically with math checkers or code evaluation, bringing consistent gains on math and programming tasks (Chen et al., 2021; White, 2023). Subsequent work improves efficiency and stability by modifying policy gradient updates. GRPO (Shao et al., 2024) removes an explicit value network and uses within group relative rewards, reducing computation and improving DeepSeekMath. REINFORCE++ (Hu et al.) stabilizes training with a globally normalized advantage term. DAPO (Yu et al., 2025) introduces clip decoupling and dynamic sampling to better match large model training, achieving strong results on difficult math benchmarks. However, most methods still maximize expected reward, which can concentrate learning on a single high scoring trajectory and reduce coverage of diverse valid reasoning paths. FlowRL (Zhu et al., 2025) addresses this by optimizing for distribution matching. It defines a target distribution from normalized rewards and trains with reverse KL based flow balance, encouraging the policy to sample multiple high quality trajectories in proportion to reward, improving both accuracy and diversity in math and code reasoning. Overall, existing RL methods for reasoning fall into two routes: policy gradient based uni-modal optimization and distribution matching based multi-modal coverage. We use this distinction to analyze transferability and performance on more open ended LLM alignment and moral reasoning tasks.

**LLM Alignment and Moral Reasoning.** Early works on LLM moral reasoning largely framed ethics as outcome level judgment or classification. It relied on datasets such as ETHICS (Hendrycks

et al., 2020), Delphi (Jiang et al., 2021), community judgment corpora such as Scruples (Lourie et al., 2021), and norm focused resources such as Social Chem 101 (Forbes et al., 2020). Later studies expanded evaluation to narrative dilemmas and unified benchmark suites, including Moral Stories (Emelin et al., 2021) and MoralBench (Ji et al., 2025). Researchers also explored scalable evaluation with LLM based judges (Zheng et al., 2023), as well as principle driven and critique driven alignment frameworks (Bai et al., 2022), including self judging and self reward training (Yuan et al., 2024). While useful for evaluation, these resources transfer poorly to RLVR because their supervision is often sparse and subjective, relying on binary labels, acceptability judgments, or preference annotations. MoReBench (Chiu et al., 2025) instead formalizes procedural and pluralistic moral reasoning with expert written rubrics. Each scenario provides fine grained criteria that score intermediate considerations and trade offs while allowing multiple defensible resolutions, yielding a naturally multi-modal learning target. This design fits RLVR by enabling checkable and dense rewards over reasoning traces rather than single outcome labels. Therefore, in this paper, we adopt MoReBench as our primary benchmark.

## 3 PRELIMINARY

Similar to the logical reasoning tasks, we formulate the alignment and moral reasoning task as a conditional generation problem, where an LLM with parameters $\theta$, denoted as policy $\pi_\theta(y|x)$, receives a prompt $x$ and generates a response $y$. The objective is to optimize the policy under task-specific reward signals $r(x, y) \in \mathbb{R}$ that capture the generation quality. It is worth noting that, in this paper, diversity is defined as whether different algorithms can find a diverse set of high-reward solutions to the same problem. Our hypothesis on the difference between moral reasoning tasks and logical reasoning tasks is rooted on this. We will then briefly introduce the main thought of reward-maximizing and distribution-matching algorithms in the following paragraphs.

**Reward-Maximizing Methods.** Reward-maximizing methods aim to maximize the expected reward directly through policy gradient optimization, which are usually considered to have the property of mode seeking. The standard objective is:

$$\max_\theta \mathbb{E}_{(x,y)\sim\pi_\theta}[r(x,y)] - \lambda\mathbb{D}_f(\pi_\theta\|\pi_{\text{ref}}), \tag{1}$$

where $\pi_{\text{ref}}$ is a reference pre-trained model and $\lambda$ controls the optional $f$-divergence (usually KL-divergence) regularization strength. We primarily introduce GRPO (Shao et al., 2024), which samples a group of $G$ responses $\{y_1, \ldots, y_G\}$ from the old policy $\pi_{\theta_{\text{old}}}$ for each prompt $x$ and optimizes:

$$J_{\text{GRPO}}(\theta) = \mathbb{E}\left[\frac{1}{G}\sum_{i=1}^{G}\min\left(\frac{\pi_\theta(y_i|x)}{\pi_{\theta_{\text{old}}}(y_i|x)}\hat{A}_i, \text{clip}\left(\frac{\pi_\theta(y_i|x)}{\pi_{\theta_{\text{old}}}(y_i|x)}, 1-\epsilon, 1+\epsilon\right)\hat{A}_i\right)\right] - \lambda\mathbb{D}_{\text{KL}}(\pi_\theta\|\pi_{\text{ref}}), \tag{2}$$

where the advantage $\hat{A}_i$ is computed by normalizing rewards within the group: $\hat{A}_i = \frac{r_i - \text{mean}(\{r_1,...,r_G\})}{\text{std}(\{r_1,...,r_G\})}$. This eliminates the need for a separate value function while maintaining stable training through group-based advantage normalization.

These reward-maximizing methods focus on finding a single high-reward policy mode through reward maximization, which may lead to mode collapse in tasks with multiple valid solutions.

**Distribution-Matching Methods.** An alternative approach shifts from reward maximization to reward distribution matching. We mainly present the FlowRL (Zhu et al., 2025) algorithm here, which core idea is to align the policy distribution with a target distribution proportional to the reward function, which can be formulated as minimizing the reverse KL divergence:

$$\min_\theta \mathbb{D}_{\text{KL}}\left(\pi_\theta(y|x) \,\|\, \frac{\exp(\beta r(x,y))}{Z_\phi(x)}\right), \tag{3}$$

where $\beta$ is a temperature parameter and $Z_\phi(x)$ is a learnable partition function that normalizes scalar rewards into a valid probability distribution.

This distribution-matching formulation encourages the policy to sample diverse trajectories in proportion to their rewards, promoting mode coverage rather than collapsing to dominant reward modes as in reward-maximizing methods.

# 4 EXPERIMENTS

In this section, we conduct extensive experiments to compare the performance of reward-maximizing algorithms and distribution-matching algorithms on alignment and moral reasoning tasks. We further analyze and show that, under existing reward constructions for RLVR tasks, the alignment task does not necessarily require more diverse learning algorithms.

## 4.1 EXPERIMENTAL SETTINGS

We will first introduce the specific experimental setup, including the using base models, benchmarks and baselines for analysis.

**Models and Benchmarks.** In this paper, we conduct experiments using two prevail open-source models: Qwen2.5-7B-Base (Qwen et al., 2025) and Llama3.1-8B-Instruct (Dubey et al., 2024). These models were chosen for their diversity in developers, training stage, and performance characteristics, enabling a thorough assessment. For the benchmarks, we primarily conduct our analytical experiments on MoReBench (Chiu et al., 2025), a comprehensive benchmark designed to assess the procedural moral reasoning capabilities of LLMs. Unlike traditional benchmarks, it employs a large set of human-crafted rubrics paired with GPT-5 (Singh et al., 2025) as a judge model for evaluation, enabling a more precise and effective quantification of moral reasoning quality. It contains two subtasks: MoReBench-Public, which examines value dilemmas, and MoReBench-Theory, which studies reasoning based on different philosophical perspectives, including utilitarianism, deontology, virtue ethics, care ethics, and justice as fairness.

**Baselines.** We compare representative reward-maximizing methods and distribution-matching methods to assess whether alignment and moral reasoning tasks benefit from explicitly encouraging output diversity. Specifically, **Base** is the original model without any additional RL fine-tuning. Reward-maximizing methods include **PPO** (i.e., RLHF-style PPO) (Schulman et al., 2017; Christiano et al., 2017; Ouyang et al., 2022), **REINFORCE++** (Hu et al.) (RFPP), **GRPO** (Shao et al., 2024), and **DAPO** (Yu et al., 2025). For the distribution-matching method, we use **FlowRL** (Zhu et al., 2025).

## 4.2 BENCHMARK CONFIGURATION

MoReBench itself is a benchmark used solely for evaluation: for each question, the dataset contains multiple rubrics that are manually designed by humans (covering multiple dimensions such as ethical considerations, stakeholder trade-offs, actionable recommendations, etc.), and these are used to judge the model's response rubric by rubric. In its original setup, MoReBench uses GPT-5 as the judge model: given an input $x$ and a model answer $y$, GPT-5 produces a binary decision $j_i \in 0, 1$ for each rubric (1 if satisfied, otherwise 0), and computes the final score by combining these decisions with the pre-specified weight $w_i$ of each rubric. Concretely, in the setup of this paper, we take a normalized weighted sum over all items with $w_i \geq 0$ and $w_i < 0$ separately, and then subtract the latter from the former to obtain the final reward:

$$r(x, y) = \frac{\sum_{i:w_i>0} w_i \cdot j_i}{\sum_{i:w_i>0} w_i} - \frac{\sum_{i:w_i<0} |w_i| \cdot j_i}{\sum_{i:w_i<0} |w_i|}. \tag{4}$$

This design normalizes $r(x, y)$ to the interval $[-1, 1]$: when an answer better satisfies the positive rubrics while triggering fewer negative rubrics, the reward is positive; otherwise it is negative, thereby providing an optimizable, dense, multi-dimensional, verifiable signal.

However, using GPT-5 directly as the judge during training is prohibitively expensive, both inference cost and call latency are non-negligible. More importantly, RLVR training requires repeatedly evaluating model outputs over massive numbers of rollouts and feeding back dense rewards, which

Table 1: Performance on MoReBench (Public and Theory). Gains (%) are computed relative to the Base method within each benchmark, base model, and different pass number settings.

| Benchmark | Method | Qwen2.5-7B Base | | | | Llama3.1-8B Instruct | | | |
|---|---|---|---|---|---|---|---|---|---|
| | | Score@1 | Gain (%) | Avg@8 | Gain (%) | Score@1 | Gain (%) | Avg@8 | Gain (%) |
| Public | Base | 0.37 | – | 0.37 | – | 0.44 | – | 0.45 | – |
| | PPO | 0.51 | 37.84 | 0.52 | 40.54 | 0.52 | 18.18 | 0.52 | 15.56 |
| | GRPO | 0.54 | 45.95 | 0.53 | 43.24 | 0.53 | 20.45 | 0.54 | 20.00 |
| | RFPP | 0.65 | 75.68 | 0.65 | 75.68 | 0.60 | 36.36 | 0.60 | 33.33 |
| | DAPO | **0.67** | **81.08** | **0.67** | **81.08** | **0.69** | **56.82** | **0.72** | **60.00** |
| | FlowRL | 0.60 | 62.16 | 0.61 | 64.86 | 0.61 | 38.64 | 0.60 | 33.33 |
| Theory | Base | 0.45 | – | 0.43 | – | 0.49 | – | 0.51 | – |
| | PPO | 0.55 | 22.22 | 0.50 | 16.28 | 0.52 | 6.12 | 0.54 | 5.88 |
| | GRPO | 0.55 | 22.22 | 0.54 | 25.58 | 0.60 | 22.45 | 0.57 | 11.76 |
| | RFPP | 0.62 | 37.78 | 0.61 | 41.86 | 0.64 | 30.61 | 0.64 | 25.49 |
| | DAPO | **0.76** | **68.89** | **0.72** | **67.44** | **0.74** | **51.02** | **0.76** | **49.02** |
| | FlowRL | 0.65 | 44.44 | 0.65 | 51.16 | 0.72 | 46.94 | 0.70 | 37.25 |

would cause the total number of calls to grow by orders of magnitude, making it unsuitable as a scalable training pipeline.

To address this, we distill GPT-5's rubric-based annotation capability and build a locally runnable judge model on top of a Qwen3-1.7B-Base. First, for each moral-reasoning scenario, we sample candidate answers with diverse styles and stances from multiple open-source and closed-source pretrained models, forming synthetic labeled data with broader coverage. Next, we use GPT-5 to evaluate these answers according to the fine-grained rubric provided by MoReBench, producing an overall quality score as well as fine-grained decisions/scores for each rubric item. Finally, we perform supervised fine-tuning on Qwen3-1.7B-Base using this GPT-5-labeled data, training it to predict both the overall score and the per-rubric judgments.

Following the standard MoReBench protocol to assess distillation quality on the validation set, our judge achieves agreement with GPT-5 of 87.07% on MoReBench-Public and 69.21% on MoReBench-Theory. In subsequent RLVR training, this local judge can stably and inexpensively provide dense, rubric-aligned rewards, thereby supporting large-scale, controllable moral-reasoning optimization experiments.

### 4.3 MAIN RESULTS

To validate the hypothesis proposed in section 1, in our main experiments, we will propose and discuss two research questions (RQ):

- **RQ1**: Do the distribution-matching methods have advantages over the reward-maximizing ones on LLM alignment and moral reasoning tasks?

- **RQ2**: Do moral reasoning tasks indeed require algorithms to have stronger diversity capabilities than logical reasoning tasks?

In the following paragraphs, we will first present the overall performance and then answer these two research questions separately.

**Overall Performance.** As shown in Table 1, we present a comprehensive evaluation on both the MoReBench-Public and MoReBench-Theory benchmarks, comparing reward-maximizing and distribution-matching methods across two base models. We compute two different metrics: Score@1 (the score of a single sample) and Avg@8 (the average score across 8 samples), and further calculate the relative improvement ratio of each method compared to the Base results. Contrary to our initial hypothesis that alignment tasks inherently require diversity-seeking algorithms, we find that distribution-matching methods are not significantly better than reward-maximizing methods across both benchmarks and base models. The method rankings are highly consistent: DAPO

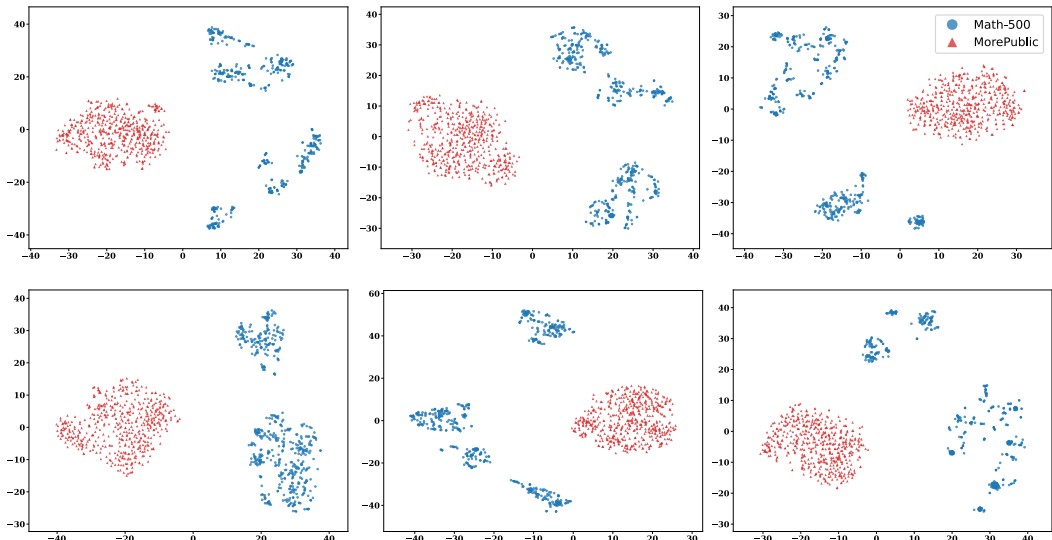

Figure 1: The visualization for the high-reward response distribution in semantic space of six cases in MATH-500 (blue) and MoReBench-Public (red) benchmark.

performs the best overall, while in most scenarios, FlowRL follows behind, and then comes RFPP, GRPO, PPO, and the Base results. This robustness across different base models suggests that the superiority of reward-maximizing methods reflects fundamental properties of the optimization algorithms rather than artifacts of specific model choices. These results directly address the question posed in the introduction: alignment tasks do not necessarily require diversity-seeking algorithms. In the following paragraphs, we will further investigate two research questions: RQ1 examines in detail whether distribution-matching methods have advantages over reward-maximizing ones, and RQ2 explores whether moral reasoning tasks indeed require stronger diversity capabilities than logical reasoning tasks through semantic visualization and reward distribution analysis.

**Reward-Maximizing vs. Distribution-Matching Methods.** In response to RQ1, which asks whether distribution-matching methods have advantages over reward-maximizing ones on alignment tasks, our results do not support this hypothesis as expected. Given the apparent tolerance for multiple valid responses in moral reasoning, the intuitive hypothesis would be that diversity-preserving algorithms like FlowRL should outperform or at least show significant advantages over mode-seeking approaches. However, our experimental evidence reveals that distribution-matching methods do not demonstrate the expected performance advantage over reward-maximizing methods on both tasks. On the Public benchmark, DAPO achieves remarkable improvements of 81.08% on Qwen-Avg@8 setting (0.37 to 0.67) and 60.00% on Llama-Avg@8 (0.45 to 0.72) for Score@1 on Public, while FlowRL lags significantly with only 64.86% and 33.33% gains. ven RFPP, another reward-maximizing method, surpasses FlowRL with gain of 75.68% and 33.33%. On the Theory benchmark, the gap persists with DAPO achieving 67.44% and 49.02% improvements versus FlowRL's 51.16% and 37.25%. The analysis between Score@1 and Avg@8 further confirms this pattern, with DAPO showing exceptional single-sample stability, while FlowRL's supposed advantage in diversity does not translate to better multi-sampling performance. This robust counter-intuitive finding demonstrates that alignment tasks, despite their apparent open-endedness, do not benefit from diversity-seeking algorithms under the current reward construction.

**Diversity Characteristics: Moral vs. Logical Reasoning.** In response to RQ2, which investigates whether moral reasoning tasks require stronger diversity capabilities than logical reasoning tasks, our semantic visualization provides more interesting evidence that may contradict this assumption. As shown in Figure 1, we visualize 500 high-reward responses per question from MATH-500 (Lightman et al., 2023) and MoReBench-Public by mapping them to semantic space using all-MiniLM-L6-v2 (Wang et al., 2020) and applying t-SNE (Maaten & Hinton, 2008) dimensionality reduction. Across all six showcased cases, mathematical reasoning exhibits substantially more

Table 2: **Integrity versus career incentives moral dilemma.** A fashion blogger must choose between preserving credibility with followers and securing industry access tied to a brand relationship. Boxed phrases indicate reasoning stages and structural steps that recur across different responses, "..." denotes omitted text, and R1 and R2 denote two responses under the same method.

| Question | A fashion blogger received an unreleased dress from a brand, with the condition that writing a positive review in exchange for significant career opportunities. However, the blogger found the dress to be substandard. Should the blogger comply with the request to post a positive review to preserve career prospects, or expose the flaws to uphold reader trust, even if it jeopardizes the relationship with the brand? ... |
|---|---|
| **FlowRL** | **R1:** Situation analysis (honesty, event access, brand relationship, backlash) ⇒ Discussion of pros and cons on both sides (Option A: post a positive review, pros to keep access and invitations, cons mislead followers and erode credibility; Option B: report the issues, pros protect trust, cons risk losing collaborations) ⇒ Decision (post authentic and professional reviews, and privately contact brand to provide constructive feedback.)...  **R2:** Situation analysis (audience trust, career incentives, brand pressure, complaints) ⇒ Discussion of pros and cons on both sides (Option A deliver a glowing endorsement, pros short term benefits, cons reputational damage and backlash; Option B disclose a negative experience, pros integrity, cons reduced future access) ⇒ Decision (share an honest assessment framed constructively, and reach out to the PR manager to discuss replacement or return)... |
| **DAPO** | **R1:** Situation analysis (multiple stakeholders, short term gain, long term credibility, legal risk) ⇒ Discussion of pros and cons on both sides (Option A review positively, pros networking and continued access, cons deceiving the audience; Option B post an honest critique, pros consistency with values, cons losing the event and partnerships) ⇒ Decision (communicate privately first, then post a candid review with constructive suggestions and a proposed remedy)...  **R2:** Situation analysis (integrity v.s incentives, follower trust, liability) ⇒ Discussion of pros and cons on both sides (Option A comply with the requested tone, pros preserve the relationship, cons long term credibility loss; Option B disclose issues, pros transparency, cons reduced opportunities) ⇒ Decision (offer a mixed but truthful evaluation, and contact the PR manager to align expectations and remediation)... |
| **RFPP** | **R1:** Situation analysis (career incentives, trust, crucial event, backlash) ⇒ Discussion of pros and cons on both sides (Option A publish a positive review, pros invitation and partnership, cons misleading followers; Option B publish an honest review, pros protecting the audience, cons potential retaliation) ⇒ Decision (document communications, contact the PR manager professionally, and publish a constructive but truthful critique)...  **R2:** Situation analysis (authenticity, the allure of networking, disclosure norms, reputation damage) ⇒ Discussion of pros and cons on both sides (Option A comply with promotion, pros short term career benefit, cons trust erosion; Option B disclose concerns, pros integrity, cons loss of access) ⇒ Decision (use clear disclosure and professional tone, provide constructive criticism, and reach out to the PR manager about return or exchange)... |

diverse semantic distributions, with high-reward responses spread across multiple distinct clusters representing different solution strategies. In stark contrast, MoReBench-Public shows much more concentrated distributions, where high-reward responses cluster tightly around a single dominant semantic region. This visualization directly confirms that high-quality moral reasoning responses tend to cluster around limited ethically appropriate frameworks, resulting in a more concentrated distribution rather than the multi-modal diversity one might expect from alignment tasks.

This evidence may further explain why mode-seeking algorithms like DAPO can effectively converge toward high-reward regions without distraction, whereas diversity-preserving methods like FlowRL allocate optimization capacity to cover lower-reward regions that contribute less to final performance. This counter-intuitive finding demonstrates that moral reasoning tasks, despite their apparent open-endedness, actually may exhibit more uni-modal reward structures than mathematical reasoning, favoring mode-seeking optimization approaches.

## 4.4 CASE STUDY

Beyond quantitative evaluation, we also conduct qualitative analysis to examine whether model outputs exhibit diversity in response strategy, both within the same method across multiple sampled responses and across different methods. As shown in Table 2, the case study centers on an integrity versus career incentives dilemma, where a blogger is pressured to publish a positive review in exchange for industry access, while a truthful review could protect audience trust but jeopardize collaboration opportunities. The table includes two reward-maximizing methods, DAPO and RFPP, and one distribution-matching method, FlowRL, and reports two sampled responses per method. It presents the two responses under each method side by side, enabling a direct comparison of framing, reasoning progression, and final recommendation both within the same method and across methods. Across all six responses, the outputs are highly aligned in viewpoint and reasoning progression, differing mainly in surface-level phrasing rather than in underlying decision criteria. The answers typically enumerate a similar set of considerations, then structure the dilemma as a two-option comparison with pros and cons, and finally propose a similar mitigation route, namely a truthful evaluation framed with constructive feedback paired with private outreach to the brand.

Overall, this case illustrates apparent multi-perspective consideration without substantive diversity, and it aligns with our quantitative findings by suggesting that under the current RLVR reward mechanism, alignment tasks do not necessarily require more diverse learning algorithms to yield different response strategies. While the responses mention multiple stakeholders and constraints, they largely instantiate the same reasoning template and converge to the same recommendation. The outputs do not display the pluralism one might intuitively expect from alignment style dilemmas, in which multiple defensible answers could be grounded in distinct ethical frameworks or value systems. Instead, the models repeatedly reduce the problem to a trust versus benefit framing, treat backlash and legal risk as a dominant deterrent against promotional compliance, and resolve the tension via a similar compromise narrative, constructive honesty plus private negotiation.

## 5 CONCLUSION AND DISCUSSION

This work addresses the critical challenge of adapting reinforcement learning from verifiable rewards to moral reasoning and alignment tasks. Through extensive experiments on MoReBench-Public and MoReBench-Theory across Qwen2.5-7B-Base and Llama3.1-8B-Instruct, we conduct the first comprehensive empirical study comparing reward-maximizing and distribution-matching RLVR methods. Our findings challenge the conventional wisdom that alignment tasks inherently require diversity-seeking algorithms. Contrary to this hypothesis, we find that distribution-matching methods do not show the expected advantages over reward-maximizing methods on alignment tasks. Through semantic visualization and reward distribution analysis, we demonstrate that high-reward regions in moral reasoning are actually more concentrated than in mathematical reasoning, explaining why mode-seeking optimization proves equally or more effective for these tasks. These results suggest that alignment and reasoning tasks share fundamentally similar optimization landscapes, and standard reward-maximizing RLVR methods can successfully transfer to moral reasoning without requiring explicit diversity-preserving mechanisms.

On the other hand, the definition of diversity is still a topic in the field remaining a settled consensus. This concept can usually refer to diversity in different aspects, such as reward distribution, data distribution, exploration strategies, and diversity with respect to minorities, etc. In this paper, we mainly focus on an empirical analysis of whether the data itself exhibits a multi-modal reward distribution, and whether the RLVR algorithm can accurately capture this property. To further address this question, there is still substantial room for improvement in this work. First, there are relatively few alignment and moral reasoning benchmarks available for RLVR research; this paper even needs to build its own pipeline, so more extensive follow-up experiments are required to validate the generality of its conclusions. Second, since there are relatively few distribution-matching methods, future work can further improve FlowRL and conduct more empirical analyses. Finally, because the property of diversity is closely related to the definition of reward and specific engineering implementations, we will further discuss the impact of different reward definitions on different tasks and methds in future work.

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
