# OpenReview forum: "Does LLM Alignment Really Need Diversity? An Empirical Study of Adapting RLVR Methods for Moral Reasoning"
_ICLR.cc/2026/Workshop/AFAA — Submitted to AFAA 2026_

### Official Review · Reviewer_h5cN · 2026-02-18
**Interesting research question with promising results.**

**Rating:** 4
**Confidence:** 3

**Summary:**

This paper investigates whether diversity-seeking algorithms are necessary for improving LLM alignment in moral reasoning tasks, compared to reward-maximizing methods. Although moral reasoning admits multiple valid responses, experiments on MoReBench across two model families and six RL methods show no significant advantage for distribution-matching approaches. Further analysis of high-reward response distributions suggests that moral reasoning exhibits more concentrated reward regions than mathematical reasoning, challenging the hypothesis that alignment tasks inherently require diversity-preserving optimization.

**Strengths:**

- Clear and well-written presentation. The paper is well structured and easy to follow. The research questions and contributions are clearly stated, and the related work provides sufficient context.
- Choice of models. The experiments are conducted on two different model families (different from the judge or not), which strengthens the empirical evaluation.
- Empirical results. Across two base models and six methods, the results show consistent trends and stability, including similar performance patterns when increasing the number of samples (e.g., from 1 to 8). This suggests the findings are robust, even though standard deviations are not provided.
- Timely and relevant research question. The work addresses an interesting and important question regarding the role of diversity in LLM alignment and moral reasoning, which is highly relevant to current research.

**Weaknesses:**

- Limited validation of the LLM-based judge. While the distilled judge is validated through agreement with GPT-5, the paper does not evaluate its alignment with human judgments. Additional validation against human annotations would strengthen the reliability of the evaluation.
- Reproducibility. The paper does not provide sufficient information about training parameters, hyperparameters, and the full reward pipeline, which makes it difficult to reproduce the experiments. More detailed implementation details would improve transparency and facilitate future work.
- Interpretation of diversity-related findings. Although FlowRL (the distribution-matching method) performs slightly worse than the best reward-maximizing method (DAPO), it consistently outperforms several other reward-maximizing baselines. Therefore, the results may indicate the need for more advanced distribution-matching algorithms rather than concluding they are not beneficial.

---

### Official Review · Reviewer_1qBj · 2026-02-20
**Review of "Does LLM Alignment Really Need Diversity? An Empirical Study of Adapting RLVR Methods for Moral Reasoning"**

**Rating:** 2
**Confidence:** 4

**Summary:**

This paper investigates whether LLM alignment and moral reasoning tasks inherently require diversity-seeking RL algorithms. The paper compares reward-maximizing methods (PPO/GRPO/DAPO) and distribution-matching methods (FlowRL).

**Strengths:**

• Research is well-motivated and provides a systematic comparison of these two RL methods (reward maximizing vs distribution matching) for moral reasoning and alignment. The paper challenges the assumption that alignment tasks inherently require diversity-seeking RL algorithms.
• Distillation of GPT-5 into a Qwen3-1.7B local judge enables scalable RLVR training for moral reasoning. This rubric-based judge model is quite practical and could even be run locally.
• Results are consistent across the model families tested of Qwen2.5-7B-Base and Llama 3.1-8B-Instruct.
• Core insight of the paper from semantic analysis is that moral reasoning high-reward regions may be more concentrated than math reasoning, explaining why mode-seeking methods like DAPO perform well.

**Weaknesses:**

• 69% agreement with GPT-5 on MoReBench-Theory subset raises concerns about the reward noise affecting conclusions.
• The core claim of this paper relies entirely on the reward setup wherein the unimodality may be a result of the rubric design and distilled judge bias rather than the underlying task structure.
• The diversity analysis simply relies on visualizations rather than offering an empirical metric or deeper quantitative testing.
• It is unclear if the conclusions will hold more generally since only one distribution-matching method of FlowRL was used and only one moral reasoning benchmark MoReBench was evaluated. Reward-maximization may lead to reward hacking or extreme mode collapse in complex and dynamic alignment scenarios. The scope is too limited to be able to extend to general claims of alignment.
• Using all-MiniLM-L6-v2 for semantic mapping may not capture the nuanced differences between the responses.

---

### Official Review · Reviewer_ebRQ · 2026-02-21
**Does LLM alignment really needs diversity?**

**Rating:** 3
**Confidence:** 4

**Summary:**

The paper has an interesting premise: whether alignment really needs diversity. Alignment task inherently requires diversity-seeking distribution-matching algorithms rather than a reward-maximizing policy methods. To investigate this, the authors conduct a study comparing both reward maximization algorithms as well as reward matching algorithms to a specific benchmark called MoReBench. They also visualize high reward responses made by few open source models on math benchmarks & MoReBench benchmarks to see the reward distribution differences. They propose that alignment tasks do not inherently require diversity-preserving algorithms, and that standard reward-maximizing RLVR can actually work for open-ended diverse tasks as well.

**Strengths:**

The paper provides an interesting study of how RLVR could potentially work or be useful for alignment tasks that usually require diversity. They look into designing a pipeline which uses MoreBench as an eval.

- They cover  different standard RLVR algorithms, such as PPO, GRPO, and DAPO, where DAPO performs better than distribution matching algorithms like Flow RL.

- They introduce a distillation pipeline to distill GPT-5 judgments to a Qwen3-1.7B-Base model, which is used for RLVR as a judge.

- They visualize reward distributions and compare how high-reward responses from models Qwen 2.5 7B and LLama 3.1 8B, fare  against benchmarks such as Math 500 and MoreBench, and how one could interpret each (projected) semantic space

**Weaknesses:**

- The paper proposes comparing diversity matching algorithms to reward matching algorithms on a very specific benchmark, which is MoreBench. To have a more quantitative approach on understanding whether the diversity matching algorithms are indeed not as effective as reward matching algorithms, we should have a more comprehensive set of benchmarks that are representative of capturing the alignment properties of the model.

- Although the authors explain the reasoning behind distilling the judge, it might be the case that the judge themselves have a warped/biased reward distribution, which would influence the downstream RLVR training as well as the reward matching investigation they do after. In this case, I'd be curious to see to a  diverse set of judges as well as a diverse set of tasks on which we compare RLVR to FlowRL.

- Apart from Flow RL, there are no other distribution-matching methods, so it seems a bit imbalanced of a comparison to have multiple reward-maximizing methods versus one.

- For visualization, mini-lm provides a proxy for the projection of their underlying distributions and cannot be trusted as an absolute truth.

- The authors do not go into detail on their choice for MoreBench. It'd be interesting to see some of the structural properties of More Bench or measure its diversity compared to other alignment datasets or benchmarks.

- Overall, the claims made by the paper are far too stronger to be justified by the experiments made.

---

### Meta-Review · Area_Chair_pEda · 2026-02-27

**Recommendation:** Reject
**Confidence:** 4

**Metareview:**

This paper investigates whether diversity-seeking algorithms are necessary for improving LLM alignment in moral reasoning tasks, compared to reward-maximizing methods. Although moral reasoning admits multiple valid responses, experiments on MoReBench across two model families and six RL methods show no significant advantage for distribution-matching approaches. They conclude that alignment tasks do not inherently require diversity-preserving algorithms, and that standard reward-maximizing RLVR can also work for open-ended, diverse tasks.

Reviewers provided feedback that will significantly improve the paper. The diversity analysis should not only rely on visualizations but also provide an empirical metric or more robust quantitative testing. Although FlowRL (the distribution-matching method) performs slightly worse than the best reward-maximizing method (DAPO), it consistently outperforms several other reward-maximizing baselines. Therefore, the results may indicate the need for more advanced distribution-matching algorithms rather than concluding that they are not beneficial. There are no error bars in the results.

I think this is an interesting paper that could yield interesting results for discussion. It is currently not ready for discussion in a workshop setting. I encourage the authors to address the reviewer's comments to improve the paper for future submissions.

---

### Decision · Program_Chairs · 2026-03-02

Reject